# The pocketome of G-protein-coupled receptors reveals previously untargeted allosteric sites

Janik B. Hedderich[1], Margherita Persechino[1], Katharina Becker[2], Franziska M. Heydenreich[3,4], Torben Gutermuth[1], Michel Bouvier[4], Moritz Bünemann[2] & Peter Kolb[1✉]

G-protein-coupled receptors do not only feature the orthosteric pockets, where most endogenous agonists bind, but also a multitude of other allosteric pockets that have come into the focus as potential binding sites for synthetic modulators. Here, to better characterise such pockets, we investigate 557 GPCR structures by exhaustively docking small molecular probes in silico and converting the ensemble of binding locations to pocket-defining volumes. Our analysis confirms all previously identified pockets and reveals nine previously untargeted sites. In order to test for the feasibility of functional modulation of receptors through binding of a ligand to such sites, we mutate residues in two sites, in two model receptors, the muscarinic acetylcholine receptor $M_3$ and $\beta_2$-adrenergic receptor. Moreover, we analyse the correlation of inter-residue contacts with the activation states of receptors and show that contact patterns closely correlating with activation indeed coincide with these sites.

[1] Department of Pharmaceutical Chemistry, Philipps-University Marburg, Marburg, Germany. [2] Department of Pharmacology & Clinical Pharmacy, Philipps-University Marburg, Marburg, Germany. [3] Department of Molecular and Cellular Physiology, Stanford University School of Medicine, Stanford, CA, USA. [4] Institute for Research in Immunology and Cancer, Department of Biochemistry and Molecular Medicine, Université de Montréal, Montréal, QC, Canada. ✉email: peter.kolb@uni-marburg.de

G-protein-coupled receptors (GPCRs) have evolved to transduce signals from the outside of a cell to the inside, thereby allowing the cell to respond to changes in its environment[1]. As a consequence of their role as transducers, GPCRs feature at least two interaction sites: one on the extracellular side, sensing the signalling agents (from photons to peptides), the other on the intracellular side, providing a place for the effector proteins to bind[2]. As the repertoires of extracellular signalling agents and intracellular effector proteins are quite limited, these sites are oftentimes conserved within a receptor subclass. This can pose a challenge to ligand and drug discovery efforts when the treatment of an ailment requires the selective targeting of a particular receptor subtype. An example of such a challenge are the $\beta_1$- and $\beta_2$-adrenergic receptors ($\beta_1$- and $\beta_2$AR), which differ only by a Phe/Tyr substitution in their orthosteric sites. Blockade of the $\beta_1$AR in heart by beta-blockers (such as bisoprolol) is desired for cardiovascular disease, but antagonising the $\beta_2$AR in lung tissue is detrimental for chronic obstructive pulmonary disease or asthma. Conversely, stimulation of the $\beta_2$AR (by e.g. salmeterol) helps asthma patients but potentially damages their heart through concomitant agonism of the $\beta_1$AR[3].

As a possible way of circumventing this challenge of highly similar pockets, the targeting of allosteric pockets is billed as a sensible alternative[4]. Due to the nature of GPCRs as bundles of seven transmembrane helices that are only relatively loosely coupled[5], one could indeed expect that a ligand binding to one of these pockets is able to modulate the response of a receptor. Moreover, it is generally claimed—but has never been shown—that these alternative pockets share lower sequence homology[4]. There are examples of individual ligands binding to non-orthosteric sites on a few receptors (e.g. refs. [6–9]), but it is currently unknown to what extent such binding sites exist across the receptorome and how different or similar they are in shape and sequence.

In this work, we therefore identify and analyse the ensemble of all discernible pockets—the pocketome—of 557 GPCR structures of 113 different receptors. We discover potential pockets by exhaustive docking of small molecular probes, taking into account the different electrostatics of the solvent-exposed and transmembrane parts of the receptors, and compare these data across all receptors. Based on class A and B1 structures in active and inactive conformations, we compute residue contacts including both backbone and side chain atoms. In doing so, we identify interhelical residue contacts crucial for an active or inactive state of both class A and class B1 GPCRs (we follow the nomenclature in IUPHAR's "Guide to Pharmacology" and refer to classes of GPCRs rather than families). We are then able to show that known and as-of-yet-untargeted (orphan) allosteric sites (abbreviated as KS and OS, respectively, in the following) contain such contacts of importance, speaking to the likelihood of their functional relevance. These computational investigations are strengthened with experimental studies of two model class A receptors, the muscarinic acetylcholine receptor $M_3$ ($M_3$R) and the $\beta_2$AR. Through mutations of two pockets that have not been targeted by a synthetic ligand before, we demonstrate that the residues forming these pockets are indeed involved in receptor activation after stimulation with an orthosteric agonist. Last, but not least, we compare the sequence similarity of the most frequently occurring pockets, thereby providing a quantitative assessment of their overall selectivity potential. This therefore represents the currently most exhaustive analysis of the GPCR pocketome, spanning receptors from classes A, B1, B2, C, D1, and F.

## Results

**Probe docking & conversion to volumes.** Our definition of a pocket is based on the computational docking of small molecules (probes; while the probes we used are probably too small to bind strongly to a receptor by themselves, they represent chemical moieties that are typical for GPCR ligands and are thus suited to investigate the details of cavities on receptors) to the surface of each GPCR structure individually. We therefore first show the results of our docking calculations and the conversion to volumes before turning to the identified hotspots (the pockets) themselves. Please note that, for our approach, we did not consider dimerisation of the 7TM bundle (as has been described for class C GPCRs), but rather docked to the individual monomers. Moreover, we treated each receptor structure as rigid. Exhaustively docking the 40 small, chemically diverse molecular probes (see Methods and Supplementary Table 1) into 557 structures from 113 distinct receptors, we obtained 1621367 poses in total (a more detailed description of the statistics is provided in the Supplementary Notes and Supplementary Fig. 1). We provide a list of all analysed structures together with the docking files as Supplementary Data 1[10].

To analyse the vast number of docked molecules in a statistical manner, we used our volumetric averaging algorithm (see Methods) in order to transform the poses of each docking into visualisable probe density maps. These maps are divided into equal volume elements, each of them giving information about how often a probe atom occupied a particular region. On average, each of the obtained maps consisted of 1000000 up to 3500000 volume elements. Since we wanted to investigate the density maps for trends across the different receptor classes, maps of individual receptors were added up for each class to yield a single map with higher populations overall.

**General distribution of pockets.** The class-specific density maps provided with this work can be visualised using Pymol (see Supplementary Data 2[10] for the grid files, template, and README) and might aid a reader with the following description. Said density maps reveal multiple contiguous regions that represent common cavities on the surface of all GPCRs analysed in this study (Fig. 1). Particularly for class A GPCRs, these pockets are distributed in a notably symmetric manner: both at the intra- and extracellular end of the 7TM bundle, pockets can be seen between each pair of adjacent helices. The density maps for the other classes are somewhat less well-defined and more scattered overall. This is owed to the lower numbers of structures and therefore poorer statistics, as individual structures—and possible deviations in them—carry a relatively higher weight than for the more numerous class A structures.

Here, we present only those pockets that we will discuss and examine in depth, whereas the rest of them is described in the Supplementary Notes. We chose to focus on three of the largest and—by our analysis—best-defined orphan sites and contrast them with an equal number of known sites, which we picked because they are clearly defined and because they host synthetic ligands. While the vast majority of sites defined by the densities is located at the outward-facing receptor portion (i.e. receptor residues in contact with the membrane), we also were able to identify regions of density inside the 7TM bundle. In each class, a large interhelical site (Interhelical Binding Site 1, IBS1) and adjacent secondary binding pockets (IBS2 and IBS3) can clearly be discerned. Whereas IBS1 represents the classic orthosteric site in class A GPCRs, it forms—together with the extra-cellular domain (ECD)—the peptide binding site in class B GPCRs. Furthermore, IBS2 and IBS3 are two known exosites in class A GPCRs. Since the orthosteric site of class C receptors is located in the extracellular Venus flytrap (VFT) domain, IBS1 is commonly referred to as an allosteric site in class C receptors. Our methodology was able to correctly depict the size and shape

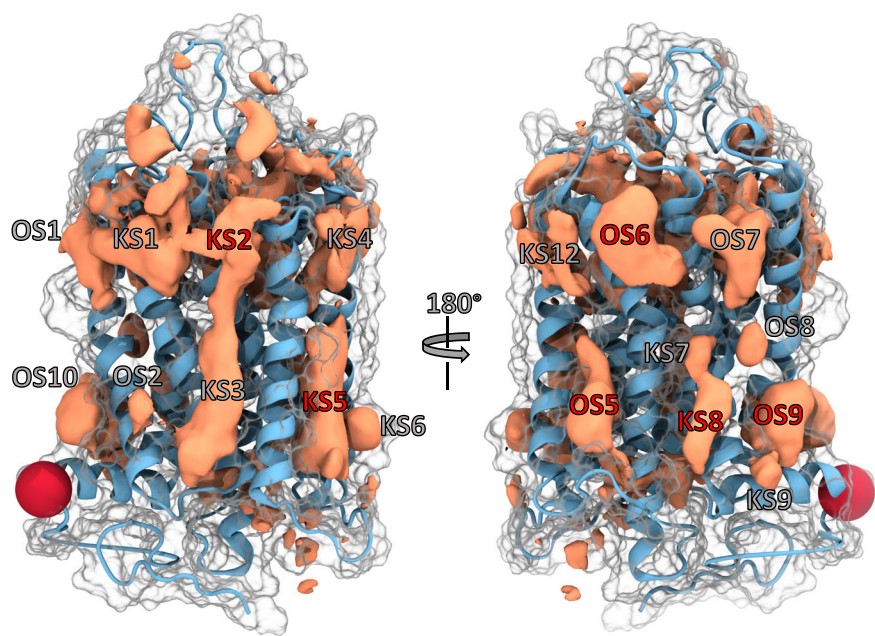

**Fig. 1 Representative depiction of the GPCR pocketome.** Cumulative densities for all class A GPCRs (orange volumes) are shown projected on the structure with PDB 1F88 ([https://doi.org/10.2210/pdb1F88/pdb]; ice blue ribbon). The surface is indicated white transparent. Visible hotspots (pockets) located at the lipid-facing receptor portion around the 7TM bundle are labelled either as OS (Orphan Site) or KS (Known Site). A more detailed description of their location is provided in the text and Table 1. We note that OS3 was described in the most recent X-ray structure PDB 7M3J[64] [https://doi.org/10.2210/pdb7M3J/pdb] during writing of this manuscript. We therefore re-labelled this site to KS12. Furthermore, OS4 was not found in the class A densities. Three known and three orphan pockets (red labels) are discussed in more detail in the text. The red sphere indicates the tip of HVIII and has been included for ease of orientation. Source data are provided as a source_data.xlsx file.

**Table 1 List of all pockets observed after probe docking, their approximate locations, and classes for which the densities are visible.**

| Site | Location[a] | Class |
|------|-------------|-------|
| OS1 | UP OF I,II | A, B1, B2, C, D1, F |
| OS2 | LP OF I,II | A, B1, B2, C, D1 |
| OS4 | MP OF V,VI | B1, B2, C, D1, F |
| OS5 | MP-LP OF V,VI | A, B1, B2, C, D1, F |
| OS6 | UP OF VI,VII | A, B1, C, F |
| OS7 | UP OF I,VII | A, B1, B2, C, D1, F |
| OS8 | MP OF I,VII | A, B1, B2, C, F |
| OS9 | LP OF I,VII,VIII | A, B1, B2, C, D1, F |
| OS10 | LP OF I,VIII | A, B1, C, F |
| IBS1 | UP-MP IF | A, B1, B2, C, D1, F |
| IBS2 | UP IF Above IBS1 IV,V, VI | A, B2, C |
| IBS3 | UP IF Next to IBS1 I,II,III,VII | A, B1, C, D1 |
| SODIUM | MP IF I,II,III,VI,VII | A, B1, C, D1, F |
| GPROT | LP IF II,III,V,VI | A, B1, B2, C, D1, F |
| KS1 | UP OF II,III | A, B1, B2, C, D1, F |
| KS2 | UP OF III,IV | A, B1, B2, C, D1, F |
| KS3 | MP-LP OF II,IV | A, B1, B2, C, D1, F |
| KS4 | UP OF IV,V | A, B1, B2, C, D1, F |
| KS5 | MP-LP OF III,IV,V | A, B1, B2, C, D1, F |
| KS6 | LP OF IV,V | A, B1, C, D1, F |
| KS7 | MP OF VI,VII | A, B1, B2, C, F |
| KS8 | LP OF VI,VII | A, B1, C, D1, F |
| KS9 | LP OF VII,VIII | A, B1, C, D1 |
| KS10 | MP IF III,V,VI | B1, B2 |
| KS11 | LP IF I,II,VII,VIII | A, B1, B2, C, D1, F |
| KS12 | UP OF V,VI | A, B1, B2, C, D1, F |

[a]LP lower portion, UP upper portion, MP middle portion, IF inward-facing, OF outward-facing
For an extended version with the structures in which they are visible, see Supplementary Table 2. We note that OS3 was described in the recent X-ray structure PDB 7M3J[64] [https://doi.org/10.2210/pdb7M3J/pdb] during writing of this manuscript. We therefore re-labelled this site to KS12. Source data are provided as a source_data.xlsx file.

of these known pockets for different classes, and we therefore hypothesized that the other pockets identified in this work can indeed also host ligands. By aligning our density maps with each other, one can see that the average IBS1 for class C receptors protrudes significantly deeper than the one of class B1, which again goes slightly deeper than the one in class A. This is perfectly consistent with experimental evidence[11]. Due to the overall higher flexibility and thus often worse resolution of extra- and intracellular loops, pockets found within these regions will not be further analysed or discussed. Comparing the densities on the outward-facing receptor portion for all analysed GPCR classes, we assigned pocket identifiers to several volumes that appeared well-defined and clearly distinct from their neighbouring densities. This facilitated later analysis and provided the means for a common orientation and discussion. However, since not only the GPCR structures themselves but also the density map shapes differ across the classes, the reader's view on whether a particular region is an individual pocket might differ from ours. That being said, our general conclusions are independent of any such small differences in definitions. The full list of pockets is presented in Table 1 and Supplementary Table 2. Going around the 7TM bundle, one can observe regions of density at the upper and lower ends between helices V and VI. These sites are referred to as KS12 and OS5, respectively. For some classes, another separated hotspot resides right between these two sites (OS4). At the lower end of the 7TM bundle, OS5 shows a large spot for classes A and F. When directly compared to class A, the density of class B1 is subdivided into multiple regions. While for classes B2 and C a small hotspot is visible, class D1 only shows some fragmented density in front of helix V.

Another larger spot is visible between helices I and VII above helix VIII for classes A, B1, C, and F (OS9). The classes B2 and D1 maps only show a small spot in this region, which might be due to the lack of helix VIII in the available structures.

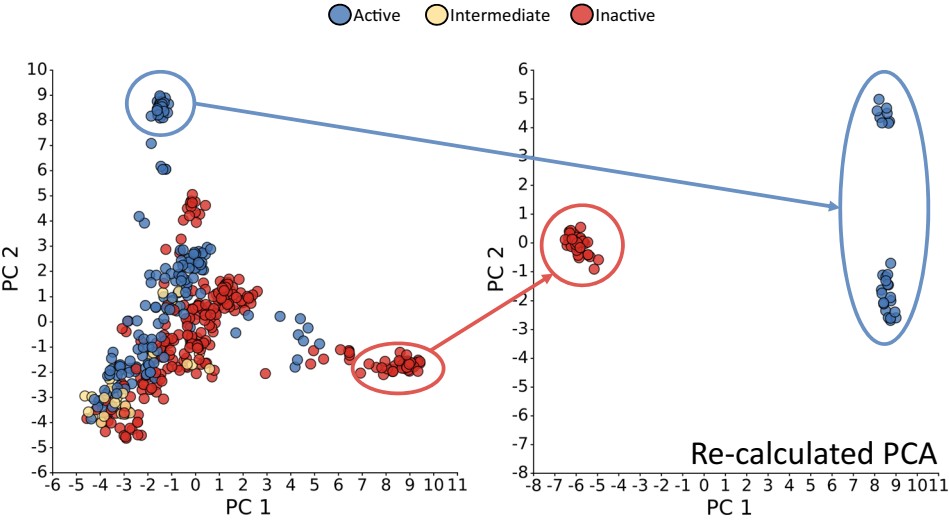

**Fig. 2 Plot of the first two principal components (PCs) of the residue contact analysis for class A structures.** Each point represents a PDB structure. They are coloured according to the GPCRdb classification into active (blue), intermediate (yellow), and inactive (red)[16]. The right panel shows the data re-calculated based on the points in the clearly separated active and inactive clusters. Each principal component (PC) value for each PDB can be seen as a linear combination of variables (i.e. contacts) that represents the residue contact landscape of a structure in a condensed manner. The first two PCs shown here explained most of the variance across all structural data, hence represent the most interesting PCs for investigating differences between receptor states on the residue contact level. Source data are provided as a source_data.xlsx file.

Encouragingly, we identified density near the region of the sodium binding pocket (SODIUM) for some classes. While classes A, B1, and F show somewhat weaker densities, the class C IBS1 extends down into this region, which makes it clearly defined.

Lastly, two regions of density were found at the intracellular portion of the 7TM bundle for all classes. Here, one spot could be identified as the G-Protein binding site between helices II, III, V and VI (GPROT). Adjacent to it, density for KS11 resides between helices I, II, VII, and VIII.

Despite the fact that we only considered monomeric subunits of the 7TM bundle in our calculations, our methodology was able to also reveal all dimerisation interfaces, which have predominantly been described for class C GPCRs. The conserved helix VI-helix VI dimerisation interface in active-state class C receptors encompasses KS7, KS8, KS9, and partially OS5 and is known to bind positive allosteric modulators (PAMs)[12,13]. Two other dimerisation interfaces can be found between helix III-helix IV (mGlu2) or helix III-helix V (GABA$_B$) in inactive-state class C GPCRs[14,15]. While the former is mainly formed by residues at the extracellular end of the helices and is thus represented by KS2, the latter dimerisation interface is located in the region of KS5.

**GPCR states can be described by their residue contact network.** In order to provide evidence that it is possible to achieve modulation of receptor function with a ligand binding to one of the allosteric pockets, we investigated to what extent these pockets are formed by residues that also participate in contact patterns specific for an active or inactive conformation of the receptor. The rationale is that residues which are involved in crucial state-specific contacts are more susceptible to interference by a ligand. In Fig. 2 and Supplementary Fig. 2, the principal component analyses of the class A and B1 residue contacts are shown, respectively. Here, we decided to focus on the first two components, since they contributed the most to the overall variance as shown in Supplementary Fig. 3 and revealed a clear separation of activation states. Across the diagonal of the PC1 vs. PC2 plot for class A, we identified a distribution of states ranging all the way from structures classified as active to those classified as inactive,

with intermediate structures positioned inbetween, congruent with the assignment of states in GPCRdb[16]. To a certain degree, the large accumulation of structures in the bottom left shows a mixture of the three classifications. Interestingly, structures classified as inactive are spread over a wide range of values of PC1, with only small differences in PC2, while active and intermediate structures display greater variance along PC2. Our contact-map-based PCA seems to indicate a slightly different view of activation compared to the assignment in GPCRdb which is based on helix II-helix VI distance cutoffs, the presence of G-protein or arrestin and further similarity measurements. The re-calculated PCA for those points that are clearly active or inactive according to our measures shows that one principal component is sufficient in order to explain the difference between the residue contacts of clearly active and inactive structures.

The PCA for class B1 contacts (Supplementary Fig. 2) shows that the structures classified as active or inactive are separated along the second principal component. Notably, four structures are separated from the others across the first principal component. As, by the time of this analysis, only one B1 structure with an assignment as an intermediate conformation in the GPCRdb was available, it was not included in the PCA. As for class A, the class B1 PCA was re-calculated considering only those structures belonging to the groups of points clearly classifiable as active or inactive. The four outliers described before were not considered in this recalculation. As expected, the PCA now shows a separation of the states across the most important first principal component.

Based on our analysis for two GPCR classes, we show that the structural state of a receptor by GPCRdb definition is closely linked to its entire residue contact network. However, we point out that a non-negligible number of class A GPCR structures with a GPCRdb-assignment as active or inactive would fall into the intermediate classification by our contact map categorisation (351 out of 417 class A structures). Hence, the residue contact map of a given structure might provide additional information on top of the GPCRdb definition of a conformational state based on interhelical distances and the type of co-crystallised ligand. Finally, we used the well-separated groups of structures from the re-calculated PCA (Fig. 2, right panel) to extract the most important and conserved active- and inactive-state-specific

contacts for each of the sites of interest. We focused on contacts formed between residues of two distinct helices, since such contacts could potentially be targeted by a ligand.

**Identification of known pockets.** As mentioned in the general description (above and Supplementary Notes), we found all the allosteric binding sites already known from crystallographic experiments (e.g. refs. [6–9]), which can be considered an excellent validation of the general applicability of our docking-based

approach (see Table 1). In this section, we focus on one exemplary site, describe its conservation across the receptorome and explain possible modes of action by using our residue contact data. Two more sites are discussed in the Supplementary Notes.

This known pocket, KS2, is located at the outward-facing residues of the upper ends of helices III and IV. While this site is only known for two class A GPCRs, namely the free fatty acid receptor FFAR1 and protease-activated receptor-2 (PAR2), our density maps show that it seems to be conserved across all GPCR classes. In order to further validate this finding, we analysed the receptorome-wide sequence identity and similarity of residues forming this site by using the definition of Table 2. While the matrices in Supplementary Figs. 4 and 5 show that the overall identity is considerably low, the similarity based on physico-chemical properties is much higher with an average value above 50% (Supplementary Fig. 6).

We then investigated the interactions of known ligands with KS2 and compared them to our residue contact analysis for class A and class B1 GPCRs. Two cases are known from the available structural data: In the case of the FFAR1 (PDB: 4PHU[17] [https://doi.org/10.2210/pdb4PHU/pdb] 5TZR[18], [https://doi.org/10.2210/pdb5TZR/pdb], 5TZY[18] [https://doi.org/10.2210/pdb5TZY/pdb]), the agonists fasiglifam and MK-8666 penetrate between the upper ends of helices III and IV coming from the inner portion of the receptor. While being anchored by polar contacts in the orthosteric region, hydrophobic interactions are dominant in KS2. A structure for the PAR2 (PDB: 5NDZ[19] [https://doi.org/10.2210/pdb5NDZ/pdb]) reveals a different mode of binding. Here, the allosteric antagonist AZ3451 stacks against the outward-facing portion of helices III and IV while only making one polar contact

**Table 2 Position of residues making up each of the sites discussed in more detail in the text and Supplementary Notes (Ballesteros-Weinstein numbering[35] was chosen for an overall better comparability). The conservation of these residues across all analysed receptors is shown in Supplementary Figs. 4 and 5. Source data are provided as a source_data.xlsx file.**

| Site | Residues |
|------|----------|
| OS5 | 5.51, 5.54, 5.55, 5.58, 5.61, 5.62, 6.35, 6.38, 6.39, 6.41, 6.42, 6.45, 6.46, 6.49 |
| OS6 | 6.47, 6.50, 6.53, 6.54, 6.57, 7.33, 7.34, 7.37, 7.38, 7.41 |
| OS9 | 1.45, 1.49, 1.52, 1.53, 1.56, 7.47, 7.50, 7.51, 7.54, 7.55, 8.48, 8.50, 8.51, 8.54 |
| KS2 | 3.23, 3.26, 3.27, 3.30, 3.31, 3.34, 4.54, 4.57, 4.58, 4.61, 4.62 |
| KS5 | 3.41, 3.44, 3.45, 3.48, 3.52, 4.41, 4.44, 4.45, 4.48, 4.52, 4.55, 5.45, 5.46, 5.49, 5.50, 5.53, 5.57 |
| KS8 | 6.35, 6.36, 6.39, 6.40, 6.42, 6.43, 7.48, 7.51, 7.52, 7.56 |

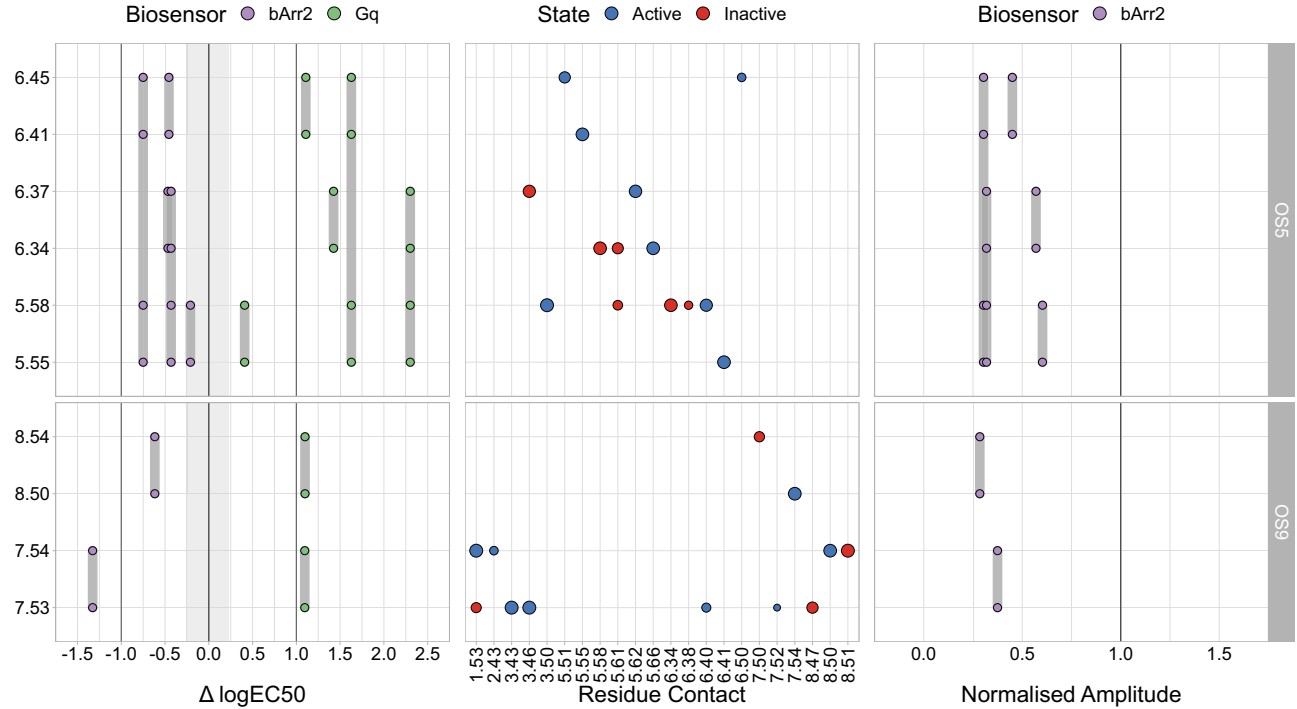

**Fig. 3 Experimental data for OS5- and OS9-mutants in the M₃R.** The y-axis depicts the position of the pocket residues (Ballesteros-Weinstein numbering[35]). Grey lines linking the points connect the respective double and quadruple mutants. The left panel shows the difference between the mutant logEC$_{50}$ and the mean wildtype (wt) logEC$_{50}$ values (logEC$_{50}^{mut}$ − logEC$_{50}^{wt}$) and the right panel the normalised amplitude (Amp$^{mut}$/Amp$^{wt}$) of the extent to which β-arrestin2 was recruited. While in the left plot a value of 0 corresponds to no potency changes, a value of 1 in the right plot corresponds to no changes in efficacy compared to wt. The greyed out area indicates minimal and maximal potency shifts of multiple G$_{αq}$ wt measurements. The central panel depicts interhelical residue contacts of each of the mutated residues that are important in active (blue) or inactive (red) conformations of class A GPCRs in a similar manner to Supplementary Fig. 10 (normalised PCA coefficient cut-off: 0.5). The point size correlates with the normalised PCA coefficient for a given contact and can be seen as a direct measurement of importance for a given state. Source data are provided as a source_data.xlsx file.

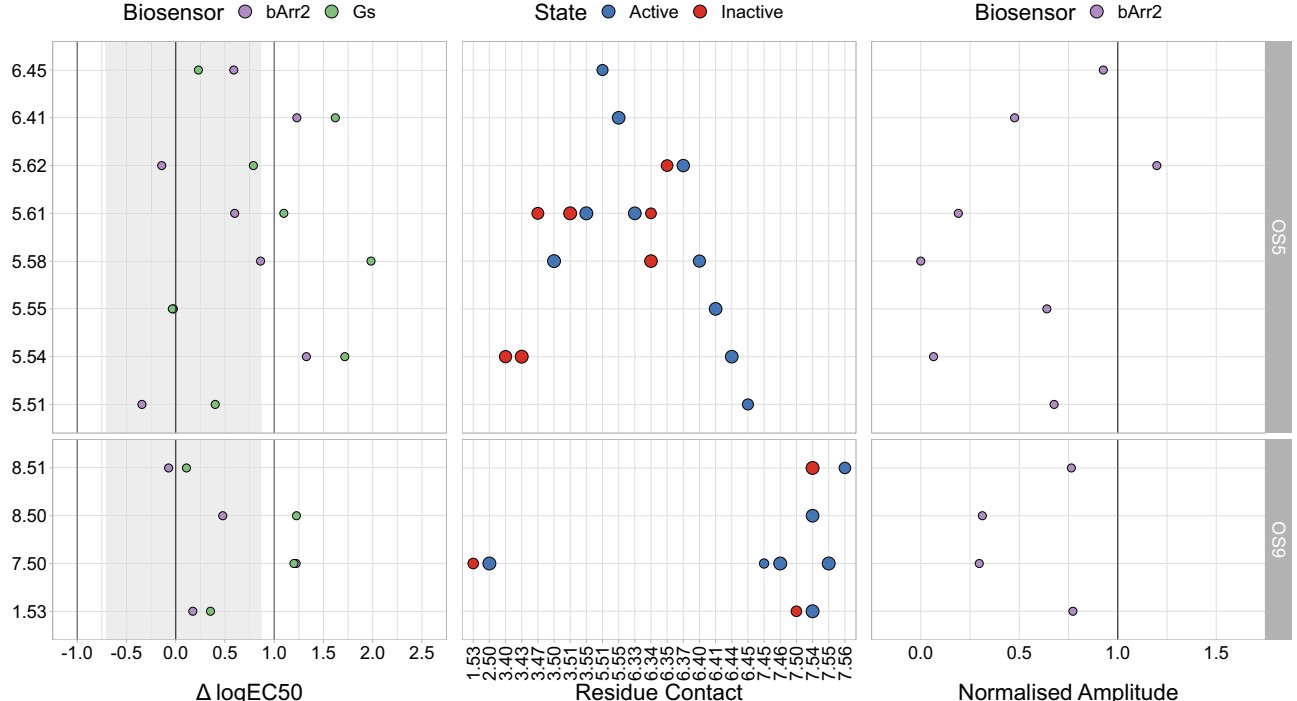

**Fig. 4 Experimental data for OS5- and OS9-mutants in the $\beta_2$AR.** The y-axis depicts the position of the pocket residues (Ballesteros-Weinstein numbering[35]). The left panel shows the difference between the mutant logEC$_{50}$ and the mean wt logEC$_{50}$ values (logEC$_{50}^{mut}$ − logEC$_{50}^{wt}$) and the right panel the normalised amplitude (Amp$^{mut}$/Amp$^{wt}$) of the extent to which $\beta$-arrestin2 was recruited. While in the left plot a value of 0 corresponds to no potency changes, a value of 1 in the right plot corresponds to no changes in efficacy compared to wt. The greyed out area indicates minimal and maximal potency shifts of multiple G$_{\alpha s}$ wt measurements. The central panel depicts interhelical residue contacts of each of the mutated residues that are important in active (blue) or inactive (red) conformations of class A GPCRs in a similar manner to Supplementary Fig. 10 (normalised PCA coefficient cut-off: 0.7). The point size correlates with the normalised PCA coefficient for a given contact and can be seen as a direct measurement of importance for a given state. Part of the G$_{\alpha s}$ data was also published in[65]. Source data are provided as a source_data.xlsx file.

to residue 3.30. Instead of pushing the two helices apart, this allosteric ligand seems to hold them together, mainly through hydrophobic interactions. Our class A contact analysis for known sites shown in Supplementary Fig. 7 revealed multiple helix III–IV contacts crucial for an inactive conformation of the receptor such as 3.23–4.61, 3.27–4.61, 3.30–4.60, and 3.34–4.58. Furthermore, this analysis indicated one highly conserved active state contact, 3.30–4.61.

**Identification of orphan pockets.** Here, we focus on two orphan sites that could represent binding sites for allosteric modulators. They are among the best-defined and largest-volume sites that emerged from our analysis based on the previously mentioned docking of small molecular probes. A third site is discussed in Supplementary Notes. Since no structural data of ligands binding to these regions is known yet, we will describe the pockets based on their amino acid sequences and our class A and class B1 residue contact analysis.

Similar to the three known sites described in more detail in this work, these pockets also reside in the outward-facing portion of the receptor. As shown in Supplementary Fig. 8, they mainly consist of hydrophobic residues. This is expected, since most of their volume lies within the membrane portion of the GPCRs. The first orphan site discussed here, OS5, is located at the lower portion of the 7TM bundle between helices V and VI. Density in this region was conserved across all GPCR classes. However, our receptorome-wide sequence similarity analysis reveals that the physicochemical properties slightly differ across the classes. While class A and B1 receptors share a high sequence similarity

with each other in this region, GPCRs belonging to classes C and F only show high sequence similarity within their respective subclass. This fact might explain the different shapes of the densities. In direct comparison to the other classes, the class F density is shifted more towards the intracellular side of the GPCR. Hence, the receptorome-wide definition of OS5 (Table 2) might not be suitable for class F receptors. In contrast, the position and shape of OS9 between helices I, VII and VIII is highly conserved across all GPCR structures. The physicochemical properties of this site are more conserved across classes A, B1 and C, with an average sequence similarity above 50%. Again, one exception is class F with a much lower homology to the other classes.

In order to identify the impact of OS5 and OS9 on the biological function of the receptors, we also conducted mutation studies with the M$_3$R and $\beta_2$AR. A visualisation of location of the residues mutated in our experiments is shown in Supplementary Fig. 9. Of note, we chose the residues such that their side chains are pointing into the sites, and are thus available for interaction with a ligand. For the M$_3$R, we constructed double or quadruple mutants, where two or four, respectively, of the residues that form these pockets were changed (residues mutated in a particular mutant are connected by a grey vertical bar in Fig. 3). These mutants were tested in a BRET-based G-protein activation assay as well as in a FRET-based $\beta$-arrestin2 recruitment assay. The summary of the resulting data is shown in Fig. 3.

Regarding OS5, our results show a clear increase in the logEC$_{50}$ values of the concentration-response curve of acetylcholine-induced G$_{\alpha q}$ activation relative to M$_3$R wt (Fig. 3 left and Supplementary Fig. 11A). Of note, except for one double mutation, all others resulted in shifts of at least one log unit. In

addition, a marked decrease in the efficacy of acetylcholine to recruit β-arrestin2 to the mutant $M_3Rs$ was observed, with only a small effect on the $logEC_{50}$ value. This is consistent with our finding that residues forming this pocket are involved in rearrangements important for receptor activation as shown in the middle panel of Fig. 3.

The same is true for OS9, where a similar right shift of $G_{\alpha q}$ activation in the mutants occurs (Fig. 3 and Supplementary Fig. 11B). Similarly, the extent of the recruitment of β-arrestin2 is substantially reduced for all mutations of OS9 (Fig. 3 and Supplementary Fig. 12), without major effects on the $logEC_{50}$ values, suggesting a major impact of the mutants on agonist efficacy. By demonstrating that the partial agonist arecoline led to a greatly diminished G-protein activation even under saturating conditions (Supplementary Fig. 11C), we confirmed that the mutations indeed strongly affect the efficacy of muscarinic agonists.

To further support our findings, we also mutated individual OS5 and OS9 residues in the $β_2AR$. The results are summarised in Fig. 4 and full curves are shown in Supplementary Fig. 13.

Similar to our findings for the $M_3R$, our results show an increase in the $logEC_{50}$ values of adrenaline-induced activation of two different biosensors ($G_{\alpha s}$ and β-arrestin2) and a decrease in efficacy relative to the $β_2AR$ wt. Our $β_2AR$ signalling results show that the $logEC_{50}$ is increased for either $G_{\alpha s}$ or β-arrestin2 or both for half the mutations across the OS5 and OS9 pockets. Similarly, for about half of the mutations, the efficacy for β-arrestin2 recruitment is reduced. As was the case for the $M_3R$, these results further strengthen the assumption that OS5 and OS9 are indeed physiologically relevant.

In order to obtain deeper insight into the possible reasons behind the interference of residues in OS5 and OS9 with GPCR activation, we compared the mutational data with our class A contact analysis (middle panel of Figs. 3 and 4). For OS5, we found that the mutated residues were frequently involved in conserved active and inactive state contacts. Starting with residue 5.54, our analysis revealed two contacts (3.43–5.54 and 5.54–6.44) important for an inactive and active state of the receptor, respectively. The decrease of function upon mutating 5.54 indicates that 5.54–6.44 might be a contact crucial for receptor activation. The same holds true for residues 5.58, 5.61, and 6.41. In our contact analysis, we found that the region formed by these residues includes numerous contacts important for an active state. While the microswitch contacts 3.50–5.58 and 5.58-6.40 are known for their importance for an active conformation of the receptor, 5.61 shows two active-state contacts with 3.55 and 6.33, respectively. Furthermore, residue 6.41 makes exactly one crucial active-state contact, namely to 5.55. Overall, our mutagenesis experiments and residue contact analysis suggest that a considerable fraction of the residues of OS5 are involved in key active-state contacts. These residues could therefore be addressed by a synthetic allosteric modulator in order to reduce receptor activity.

A largely similar picture holds true for OS9. Here, mutating the conserved residues 7.50 and 8.50 led to a decrease in receptor function. Per our contact analysis, both residues are involved in several important active-state contacts such as 2.50–7.50 and 7.54–8.50. Finally, we mutated two residues (6.34 and 6.37) that are not part of the two sites described here by themselves, but are involved in relevant active-state contacts to OS5 and OS9 residues, namely 6.34–5.66 and 5.62–6.37. These mutations led to a decrease in $G_{\alpha q}$ activation in the $M_3R$, speaking to the occurrence of a second-shell effect.

**Occupancy of known allosteric pockets.** In order to obtain deeper insight into the properties of the allosteric pockets, we

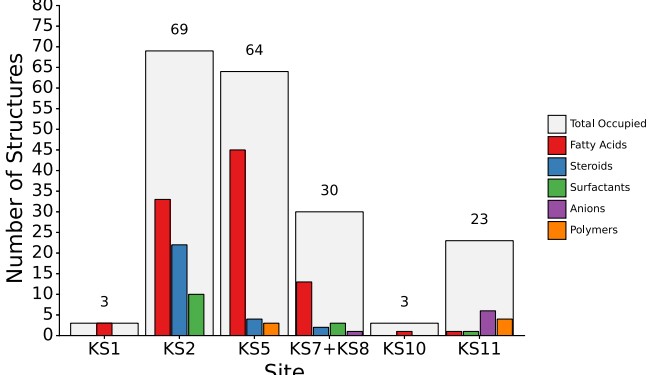

**Fig. 5 Absolute number of observed instances of components other than synthetic ligands in the Known Sites (KS).** Shown are the total number of structures in which at least one component was resolved in each site (light grey bars) and the respective numbers for surfactants (green), anions (purple), fatty acids (red), steroids (blue), and polymers (orange). Source data are provided as a source_data.xlsx file.

analysed the known sites for the occurrence of molecules besides (synthetic) ligands designed for them, i.e. focusing on crystallisation additives and co-purified substances. We tabulated all structure determination adjuvants resolved in any of the 557 structures investigated in this work (Supplementary Table 3).

This analysis revealed that in around 35% of the investigated structures, the KS host additional chemical compounds besides the added orthosteric and allosteric ligands. In particular, a recurrence of cholesterol (or cholesteryl hemisuccinate), oleic acid and glyceryl monooleate can be noticed, all of them adjuvants in purification and/or structure determination processes. Binding is more frequent to the more superficial pockets. To avoid bias introduced by the fact that not all adjuvants are present in all buffers, we also calculated the background distribution for all substances.

We found that for 12.4% (69 out of 557) of the structures, the known pockets are occupied by at least one type of crystallisation adjuvant. We grouped the additives into five categories: surfactants, steroids (cholesterol and derivatives), fatty acids, polymers (predominantly PEG and PPG), and anions. This allows us to deduce a preference of the pockets for certain types of components. Of course, not every known pocket was occupied by a component, and, as mentioned above, only a relatively small set of structures out of the total 557 contained at least one occupied pocket. Hence, we assume that the presence of a particular chemical moiety can be interpreted as a preference of a pocket rather than a random occurrence attributable to the crystallisation conditions. In other words, if a component was stable enough in order to be resolved in a pocket (and given that the electron density was sufficient to determine this), it constitutes a binding event.

A more detailed analysis of the chart (Fig. 5) elucidates a rather clear picture: The preference of KS11 for anions (26%, or 6 out of 23) is evident, probably fuelled by the interactions with R3.50 in this pocket. The next most frequent category are polymers (17.4% or 4 out of 23). In contrast, KS5 hosts an abundance of fatty acids, with approximately 70% occurence (45 out of 64), followed by the substantially smaller percentage of ≈6% (4 out of 64) of steroids.

Known pockets KS7 and KS8 are also mostly populated by fatty acids, but less frequently than in the previous case, as they are only observed up to ≈43% (or 13 out of 30). In both cases, the preference for this category is favoured by the presence of several aromatic and hydrophobic residues in the middle sections of helix III and helix IV. KS2 is the pocket with the highest value of

occupancy overall (12.4% or 69 out of 557), and in this case as well, the most recurrent category are fatty acids (≈48% or 33 out of 69), followed by steroids (≈32% or 22 out of 69). KS10 and KS1 are populated the least, only 3 structures each (i.e. less than 1%) contain a component—fatty acids in both cases (33% or 1 out of 3 and 100% or 3 out of 3 of the occupied structures, respectively).

## Discussion

In our study, we have determined the occurrence of pockets across the largest part of the currently available structural G-protein-coupled receptorome. Because the ultimate goal of this research is to identify ligands for these pockets through which receptors might be modulated in their activity, we chose the docking of small molecular probes, i.e. chemically valid compounds, as a method, rather than definitions of a pocket based on the protein surface. Thus, we optimised our definition towards what a potential ligand would see. As mentioned before, the probes by themselves are likely too small to bind with reasonable affinity. However, one can certainly imagine that connections of individual probes with appropriate linkers will lead to higher-affinity ligands, a future direction of research. We explicitly accounted for the different environments of the various receptor portions, viz. the membrane-embedded core and the solvent-exposed ends, by using a docking method where the dielectric constant of the environment can be set to the appropriate values, thus avoiding artefacts. A specifically adapted aggregation method allowed us to average over the probe docking calculations to all 557 GPCR structures. Hence, the pockets we identified exist in the majority of receptors and the pocketome we present in Fig. 1 thus constitutes a representation of the shape and distribution of frequently observable cavities. In addition, we were able to make statements on the conservation of pockets across receptor classes. Our findings strongly suggest that while the pockets are quite dissimilar in sequence space, they are more similar than generally assumed when focusing on the properties of the amino acids rather than their identity. This will merit attention when designing selective ligands, so as not to rely too much on nondirectional interactions. Despite this overall conservation, we can show that some sites are more conserved regarding their shape and physicochemical properties than others. This is also borne out at the level of the presence of crystallisation additives in the pockets. Of course, our analysis is based on rigid receptor structures and it stands to reason that some of these sites will undergo rearrangements upon changes of receptor conformation. This will need appropriate attention during ligand design. Still, the physicochemical characteristics should be exploitable for the design of class- or type-specific allosteric modulators, which has implications when designing ligands for such pockets in order to avoid unintended polypharmacology.

Most of the sites identified in this work are located on the outward-facing portion of the 7TM bundle within the membrane, and water molecules close to an allosteric ligand were only observed in 14 of the 71 structures that featured an allosteric ligand and therefore not investigated further. It cannot be ruled out that they play a role in certain sites or for certain ligands, but this is likely more important for sites that are directly accessible by the solvent such as the G-protein binding site or KS11.

We are convinced that the number of structures investigated is such that the general trends observed, at least for class A and class B1, will hold even as new structures become available. In fact, we reran our analysis pipeline shortly before submission, approximately nine months after the first time, and did not observe noticeable changes in the pocket definitions. During this time, the number of class A and B1 structures available increased from 404 to 455 and 46 to 55, respectively.

While we found all pockets that have previously been localised through structure determination with a ligand, we also identified several that have not yet successfully been targeted, at least according to publicly available data. To demonstrate that the two most prominent orphan pockets OS5 and OS9 have potential as target sites for small-molecule modulators, we mutated several of the residues lining these two pockets in two model GPCRs, the $M_3R$ and the $\beta_2AR$. In both cases, mutations had a robust effect on both G-protein activation as well as $\beta$-arrestin recruitment. Of note, we did not only observe substantial right-shifts of up to more than 10-fold of the concentration-response curves for G-protein activation (indicating the need for higher orthosteric agonist concentrations to achieve similar levels of stimulation), but also decreases in the maximum level of response for $\beta$-arrestin. These results indeed point towards the modulatory potential of ligands binding at these sites. Moreover, given the fact that $logEC_{50}$-values were shifted in opposite directions for $G_{\alpha q}$ and $\beta$-arrestin2 at the $M_3R$, one might speculate that pathway-selective ligands could be designed for these pockets. A receptor region similar to OS9 has also recently been investigated in the angiotensin II type 1 receptor[20], where it has been termed a cryptic site. As mutations of amino acids led to changes in $G_{\alpha q}$ and $\beta$-arrestin2 responses similar to what we show for our receptors, this lends further credibility to our finding that OS9 is a pan-class A pocket.

In addition, we compared the experimental findings to our contact analysis, which is based on the residue contact maps of 557 GPCR structures. We found that ligands addressing the two pockets could potentially act as negative allosteric modulators (NAMs) by disrupting contacts crucial for an active state. This was expected, since both OS5 and OS9 reside near the G-protein binding site, which is known for undergoing profound rearrangements upon receptor activation. However, it also seems plausible that by strategically targeting the inactive-state contacts or stabilising active-state contacts, one could potentially design positive allosteric modulators (PAMs) for both pockets.

Our probe docking allows us to make observations also for several of the known pockets. In the case of KS2, the available structural data from the FFAR1 and the PAR2 suggests that either positive or negative modulation of agonism at a receptor could be achieved in this pocket by separating or keeping in place, respectively, the upper ends of helices III and IV with a small molecule ligand. This hypothesis is supported by our contact analysis shown in Supplementary Fig. 7. KS2 contains highly conserved inactive-state contacts between helices III and IV, e.g. 3.23–4.61, 3.27–4.61, 3.30–4.60, and 3.34–4.58, and one crucial active-state contact, 3.30–4.61. By either breaking these contacts or keeping them intact with hydrophobic interactions, an allosteric ligand binding to KS2 could modulate the activation state of a GPCR. A hypothesis for KS5 can be found in the Supplementary Discussion.

Similar rationales as for KS2 above can be applied to the design of ligands for the orphan sites. Of course, in these cases, the challenges will be to first demonstrate that each orphan site (beyond OS5 and OS9) can indeed be exploited to modulate receptor function by small-molecule ligands, to unequivocally determine the binding locations of such ligands, and to design assays that are fast yet precise enough to be utilised in their optimisation. We certainly hope that the three-dimensional atlas laid out in this work will aid the community in achieving this goal.

## Methods

Unless stated otherwise, all operations in this workflow were scripted using python 3.7 and bash. The python packages `requests` (version 2.25)[21] and `urllib3` (version 1.25.11) were used in order to access the REST API of websites listed

below. The retrieved data was handled using `pandas` (version 1.1.4)[22]. All protein structures and sequence data were handled in `Biopython` (version 1.78)[23] and `BioPandas` (version 0.2.7)[24]. Mathematical operations were carried out using `NumPy` (version 1.19.4)[25]. Open-source PyMOL (version 2.3.0)[26] and Visual Molecular Dynamics (VMD, version 1.9.3)[27] were used for the visualisation and further editing of protein structures and volumes. Any other type of data was visualised using `plotnine` (version 0.8.0)[28] and RStudio 1.4.1717.[29]

**Collection of structural information**. Information about all available GPCR structures was fetched from the GPCRdb[30], UniProt[31], and the Protein Data Bank[32] by using our information retrieval pipeline[33]. The data most relevant for our work was extracted from GPCRdb and included the PDB identification code, UniProt entry name, class, activation state and preferred chain of each GPCR structure. This data was enriched by fetching the accession numbers and canonical amino acid sequences from UniProt. In order to correctly assign solvent-accessible and intra-membrane regions at a later stage of the workflow, information about positioning of GPCR residues relative to the membrane (inside or outside) was also included. For a more convenient and uniform handling of GPCR amino acid sequences, the canonical amino acid sequences were mapped to their respective class-specific GPCR numbering scheme. By accessing the GPCRdb generic residue number tables[34], the Ballesteros-Weinstein[35], Wootten[36], Pin[37], Wang,[38] and fungal numbering schemes were utilised for class A, B, C, F and D1 GPCRs, respectively. Finally, the PDB-formatted structures were retrieved from the Protein Data Bank.

**Preparation of structures for docking**. For each structure, the transmembrane portion and adjacent motifs belonging to the GPCR were separated from all non-native insertions (i.e. non-GPCR proteins, water, other small molecules) by using the information about the preferred chain retrieved from the GPCRdb and the DBREF tag in the PDB file. In the case of dimeric GPCR structures, only one of the monomeric subunits was considered for docking. Residues listed in the SEQADV section as expression tags and insertions were not considered. For residues that were resolved in multiple conformations, only the first conformation was extracted. Structures that contained a faulty or non-uniform DBREF or SEQADV section were manually corrected before extraction. After visually inspecting the extracted portions, the structures were prepared by using the Molecular Operating Environment (MOE, version 2020.09) software[39]. Here, incomplete residues were built utilising the "Structure Preparation" function. Termini and chain breaks that contained only one atom were removed. The built-in method "Protonate3D" was used to assign protonation states to histidine and cysteine residues. For consistency, all other residues were assigned their most frequent protonation state under physiological conditions.

Preparations were continued using CHARMM together with the CHARMM36 protein force field[40]. Termini and breaks were capped by adding ACE and NME caps to the N- and C-terminal ends, respectively. Hydrogen atoms were placed with the HBUILD command. In order to remove too close van-der-Waals contacts, an energy minimisation was carried out for each structure with a short 20-step steepest-descent optimisation followed by an adopted-basis Newton-Raphson optimisation until convergence. In order to keep as much original structural information as possible, only the side chains of formerly incomplete residues and the backbone and caps of terminal residues were allowed to move. Hydrogen atoms were rebuilt using HBUILD again after all previous operations. Then, structures were aligned by using the "cealign" algorithm as implemented in Pymol. Finally, structures were converted to MOL2 file format using UCSF Chimera[41]. The correct CHARMM atom types and charges were reassigned based on the information from the CHARMM PSF output file.

**Preparation of molecular probes for docking**. In order to exhaustively scan the receptors for possible binding sites, a diverse set of small molecular probes was assembled. Diversity was achieved by including probes with different physico-chemical properties such as size, charge and hydrogen bond acceptor/donor distribution. Forty probes were selected as representatives of different functional groups (Supplementary Table 1) and their protonation states were calculated at physiological condition using the ChemAxon Software Solution (Calculator Plugins, Marvin 20.10)[42]. MOL2 3D-conformers were generated with OpenEye's OMEGA2 and default settings[43]. Next, CGenFF4.0 parameters were generated for each probe by using the CGenFF webservice accessible via https://cgenff.umaryland.edu/[44]. In order to update the MOL2 files with the CGenFF parameters and prepare a SEED 4.1.2-ready library, scripts from the SEED 4.1.2 repository[45] were used.

**Docking calculations with SEED**. For each structure, two docking calculations were carried out using SEED 4.1.2[45]. For the first docking calculation, only the intramembranous residues were considered and the dielectric constant of the surrounding medium was set to 3.0 in order to better reflect the lipid bilayer. The second docking calculation only considered the solvent-accessible residues and the solvent dielectric constant was set to 78.5, the value for water. The SEED search algorithm works by exhaustively matching multiple copies of each molecular probe to the polar and apolar portions of the defined region, treating the protein as rigid.

The poses are then spatially clustered and evaluated with energy models that also account for receptor and fragment desolvation. The maximum number of allowed clusters per probe was set to 2000 and only the best-ranked pose per cluster was considered for the output. All other parameters and settings were used with their default values.

**Extraction of molecular features**. In order to aggregate and average the information from the SEED[45] docking calculations to volumes, a custom software was developed and applied[46]. Within this tool, docking poses are searched for substructures relevant for protein:ligand interactions using RDKit 2020.09.1.0[47] and the cartesian coordinates, atom types, molecule identity, and substructure are stored. The substructures are hydrogen bond donors, hydrogen bond acceptors, aromatic atoms, halogen atoms, basic substructures, acidic substructures, aliphatic rings and an everything substructure (SMARTS are listed in Supplementary Table 4). The docking poses output by SEED were used to construct three-dimensional grids of a user-specified voxel spacing $s_v$ (0.5 Å in this work), encompassing all molecules. For each substructure investigated, a separate grid was constructed. To reduce the influence of arbitrary parameters such as the precise grid placement and voxel boundaries, each occurrence of a substructure was not only recorded in the grid voxel it was directly located in, but also—with a fractional value—in neighbouring voxels. A distance-dependent dampening factor ensured that the majority of the change introduced in the grid is still recorded close to the grid voxel the substructure was primarily located in. In practice, each recording operation will affect four different types of grid voxels: The centre voxel (in which the substructure is located); six directly adjacent voxels sharing a surface with the centre voxel, at a distance $d$ of $s_v$; twelve voxels at $d = \sqrt{2}s_v$; and eight voxels at $d = \sqrt{3}s_v$. In each grid voxel of a type, an equal change $v$ is introduced, which is multiplied by a dampening factor $t$ and a distance penalty of $1/d$ — except for the centre voxel, where no dampening factor is applied. The change $v$ is chosen such that the overall change introduced in the grid is equal to 1, i.e. $\sum_{\text{voxels}} v \cdot t \cdot 1/d = 1$. In the present work, the variables used led to 83.34% of each change being applied to the neighbouring voxels and 16.66% to the centre voxel. Using this data, it is possible to average and visualise the areas in which each feature is frequently represented for any number of docking calculations. The grids can either be exported as a PDB file containing dummy atoms at the voxel centres that correspond to a user-given percentage of the sum of each grid or by exporting a grid file using Grid-DataFormats (https://griddataformats.readthedocs.io/en/latest/gridData/formats.html) which can be opened in commonly used molecular visualisation tools. It is also possible to calculate and save the grids of a single docking calculation and then combine multiple grids. The potential problem of grids that are not aligned is solved by constructing a master grid that is encompassing all individual grids. The values of grid voxels in the single grids are then added to the master grid using the volume overlap of the grid voxel in the eight respective grid voxels of the master grid. In this way, we were able to calculate average grids and volumes across arbitrary combinations of structures, e.g. for each GPCR class.

**Definition of allosteric pockets**. In the following, our approach of obtaining a generalised, receptorome-wide definition for each site discussed in this work is described. First, reference structures were selected for each class (A: 1F88 [https://doi.org/10.2210/pdb1F88/pdb], B1: 5EE7 [https://doi.org/10.2210/pdb5EE7/pdb], C: 7CA3 [https://doi.org/10.2210/pdb7CA3/pdb], F: 4JKV [https://doi.org/10.2210/pdb4JKV/pdb]). Since class B2 and D1 structures were heavily underrepresented in our data set, they were not considered for this generalised definition. In addition to our density maps, all structures that have an allosteric ligand were visualised. Then, all structures and maps were aligned to rhodopsin (PDB: 1F88 [https://doi.org/10.2210/pdb1F88/pdb]). The region around each density and allosteric ligand was examined and matched with the residues of the reference structures. For better comparability and in order to obtain a receptor-wide definition, the site definitions for each class were converted to Ballesteros-Weinstein numbers[35] using the GPCRdb residue tables. For each site, only the residues that occurred for at least two classes were used for the final definition.

**Sequence analysis**. The amino acid sequence of known and orphan allosteric pockets described here was analysed across the receptorome in order to determine the degree of conservation of these pockets in the GPCR spectrum. Only sequences of receptors that are structurally resolved were taken into account. For each of the sites discussed in this work, the amino acid sequence was extracted for each receptor by using the site definitions described above and the GPCRdb residue tables. For each pair of receptors, the sequence identity and sequence similarity were calculated. In order to determine the sequence similarity, the following classifications were used: polar, apolar, positively or negatively charged, aromatic. Furthermore, the overall site polarity was calculated by averaging the ratio of polar and apolar amino acids of each pocket across all receptors analysed.

**Occupancy of known allosteric pockets**. In order to verify whether the known allosteric sites identified here are also occupied by other types of compounds that could influence our results, e.g. crystallisation additives, an alignment of all 557 investigated GPCR structures was performed, using rhodopsin as the main template. Every binding site occupied by a known allosteric ligand was visually

inspected. The occurrence of the different components in the selected known pockets was then collected, grouped, and analysed. A text-based analysis of the crystallisation conditions stated in all pdb files was also performed to retrieve the background distribution and use it as reference.

**Materials for the $M_3R$**. DMEM, penicillin/streptomycin, FCS, L-glutamine, PBS and trypsin-EDTA were purchased from Capricorn Scientific GmbH, Ebsdorfergrund, Germany. Poly-L-Lysine hydrobromide, PEI and acetylcholine iodide were acquired from Sigma-Aldrich, Merck KGaA, Darmstadt, Germany. Arecoline hydrobromide was purchased from TCI Chemicals, Eschborn, Germany. Coelenterazine h was obtained from NanoLight Technologies, Pinetop, USA.

**Plasmids for the $M_3R$**. cDNAs encoding $G_{\alpha q}$-YFP[48], $G\beta1$[49], GRK2[50], $\beta$-arrestin2-mTurq[51], and $M_3$-mCit[52] were described previously. The human $M_3R$ was obtained from the Missouri S&T cDNA Resource Center. The DNA for pNluc-$G\gamma2$ was a kind gift from Dr. N. Lambert (Augusta University, Georgia, USA). The cDNA encoding mCit-$\beta$-arrestin2-mTurq was analogously cloned as described for similar reference constructs in Dorsch et al.[53] The $M_3R$ mutants were generated from these plasmids by mutagenesis using the following primers listed in Supplementary Table 5. The $M_3Rs$ containing four mutations were cloned analogously in two steps. The $M_3R$-mCit mutants were generated in the same way.

**Cell culture and transfection, $M_3R$**. All experiments were performed in HEK293T cells. Cells were cultured at 37 °C and 5% $CO_2$ in Dulbecco's Modified Eagle's Medium (4.5 g/L glucose), supplemented with 100 units/mL penicillin, 0.1 mg/mL streptomycin, 2 mM L-glutamine and 10% FCS. The cells were transiently transfected in a 6 cm dish using linear polyethylenimine (PEI) 25 kDa as the transfecting agent. For the $G_{\alpha q}$ activation experiments, HEK293T cells were transfected with the following quantities of plasmids encoding for the respective proteins: 1.5 µg $M_3R$ wt/$M_3R$ mutants, 2.4 µg $G_{\alpha q}$-YFP, 0.75 µg $G_{\beta 1}$, 0.75 µg GRK2, and 0.3 µg pNluc-$G_{\gamma 2}$. For $\beta$-arrestin2 recruitment, the cells were transfected with the following quantities of plasmids encoding for the respective proteins: 1.5 µg $M_3R$-mCit/$M_3R$-mCit mutants, 1.5 µg $\beta$-arrestin2-mTurq, and 0.75 µg GRK2. For the quantification of relative expression levels 1.5 µg of plasmid encoding for mCit-$\beta_2AR$-mTurq were transfected. The ratio of DNA and PEI was determined as 1 to 3. For 1 µg DNA, 50 µL DMEM w/o FCS were added to the DNA and PEI solutions. Both solutions were mixed, incubated at 20 °C for 30 min, being protected against light and were afterwards added to the HEK293T cells in a 6 cm dish. For the BRET-based $G_{\alpha q}$ activation, cells were counted after 24 h and 16000 cells/well were seeded into a poly-L-lysine coated 96-well plate (Greiner 96 Flat White). For the FRET-based $\beta$-arrestin2 recruitment the cells were plated into six-well plates with poly-L-lysine coated 25 mm coverslips 24 h after transfection. All experiments were performed 48 h after transfection at room temperature.

**BRET-based measurements of the $M_3R$**. Transiently transfected adherent HEK293T cells were measured in a 96-well plate with a Spark 20M Multimode Microplate Reader (Tecan), using the luciferase reporter Nluc[54]. $G_{\alpha q}$ activation was assessed with $G_{\alpha q}$-YFP/$G_{\beta 1}$/pNluc-$G_{\gamma 2}$ biosensors in the presence of $M_3R$ wt/$M_3R$ mutants[55]. Fluorescence and luminescence intensities were acquired using the Spark-Control application and the BRET emission ratio was calculated as the YFP signal (light emission between 520 nm and 700 nm) divided by the Nluc signal (light emission between 415 nm and 485 nm). In general, sixteen wells were measured in one round. Cells were washed once with extracellular buffer (137 mM NaCl, 5.4 mM KCl, 2 mM $CaCl_2$, 1 mM $MgCl_2$, 10 mM HEPES, pH 7.3) and 80 µL of a 3.07 µM solution of Coelenterazine h in buffer were added to every well. After 10 min of incubation, 10 cycles of baseline measurement were performed with a duration of about 6.5 min altogether. The measurement was paused shortly, 20 µL buffer or agonist in buffer were added and 10 cycles of agonist measurement were performed. Afterwards 20 µL agonist in a saturating concentration was added and 10 cycles of BRET measurement were performed once again. The agonist-induced change in BRET emission ratio was calculated as the difference in average values of the third cycles before and after adding the agonist. The additional change in BRET emission ratio induced by a saturating concentration of acetylcholine was calculated as the difference in average values of the third cycles before and after adding the saturating concentration of acetylcholine. The maximum change in BRET emission ratio was calculated as the sum of the agonist-induced change and the additional change induced by the saturating concentration of acetylcholine. The agonist-induced change was normalised to the maximum change in BRET emission ratio for every well. Concentration response curves were fitted by GraphPad Prism 8.3 with variable slopes. The bottom constrained to 0 and the top to 1 were determined for the concentration response curves of acetylcholine. In order to calculate the cut-off area shown in Fig. 3, all single concentration response curves of $M_3R$ wt were plotted individually and the minimal and maximal $EC_{50}$ values of wt measurements were identified.

**Single-cell FRET imaging of the $M_3R$**. A FRET-based assay was used to measure the agonist-induced interaction between $\beta$-arrestin2-mTurq and $M_3R$-mCit wt/mutant sensors[56]. The measurements were performed as previously described by Milde et al.,[57] except where declared otherwise, using an inverted fluorescence

microscope (Eclipse Ti, Nikon, Germany). The cells were excited with an LED excitation system (pE-2; CoolLED, UK) at 425 nm and 500 nm. The intensity of both LEDs was set to 2%. The fluorescence intensity was measured using the software NIS-Elements advanced research (Nikon Corporation) and the image recording frequency was set to 2 Hz. FRET emission ratio was calculated as the ratio of mCitrine intensity divided by mTurquoise intensity upon excitation of mTurquoise at 425 nm by plotting over time. All fluorescence data were corrected for background fluorescence, bleed-through and false mCitrine excitation using Excel 2019. The measurements were additionally baseline-corrected for photobleaching, using OriginPro 2018 (Originlab, USA). The cells were constantly superfused with either extracellular buffer (described in BRET-based Measurement) or acetylcholine. Every cell was stimulated for 30 s with each concentration of acetylcholine. The concentration-dependent change in FRET emission ratio were calculated as the average value of the last 5 s of stimulating with each concentration of acetylcholine. Concentration response curves were fitted by GraphPad Prism 8.3 with variable slopes.

**Quantification and correction of relative expression levels for the experiments with $M_3R$**. The relative expression level of $M_3R$-mCit and $\beta$-arrestin2-mTurquoise was corrected, using the construct mCit-$\beta$-arrestin2-mTurq for calibration of the stoichiometry. mTurquoise was excited with 425 nm whereas mCitrine was excited with 500 nm. The fluorescence intensity was measured and corrected for background fluorescence. The calibration factor was calculated as $F_{mCitrine}/F_{mTurquoise}$. For each single-cell measurement, the factor was calculated in the same way. Due to the influence on the extent of FRET signal, the relative expression level of $M_3R$-mCit and $\beta$-arrestin2-mTurquoise was corrected for an equal stoichiometry. Therefore, the factor of every single-cell FRET-measurement was divided by the calibration factor (Supplementary Fig. 16) and every measurement was multiplied with its individual reciprocal (Supplementary Fig. 12).

**Plasmids and mutagenesis for the $\beta_2AR$**. Human $\beta_2AR$ (ADRB2 except for R16 and Q27) and all biosensor constructs were assembled in pcDNA3.1. The $\beta_2AR$ was codon-optimised and a sequence encoding a SNAP tag and an N-terminal signal sequence were cloned in at the N-terminus. The biosensor plasmids are based on genes encoding Renilla luciferase (RlucII) and a GFP, either GFP10 or Renilla GFP (rGFP), with the RlucII on the $G_{\alpha s}$ or $\beta$-arrestin2, respectively, and the GFP on $G_{\gamma 1}$ and a membrane anchor (CAAX), respectively[58–60]. Wild-type $G_{\beta 1}$ was used for the G-protein activation assay. Single-point mutants of the $\beta_2AR$ were generated as described in an earlier work[61], using the primers listed in Supplementary Table 5. Non-alanine amino acids were mutated to alanine, native alanine residues were mutated to glycine. Primers were designed using custom software[62] (available at: https://github.com/dmitryveprintsev/AAScan).

**BRET-based Signalling Assays of the $\beta_2AR$**. All assays were using human embryonic kidney (HEK)-293 SL cells (a gift from Stephane Laporte). Cells were grown at 37 °C with 5% $CO_2$ in DMEM with 4.5 g/L glucose, L-glutamine, and 10% newborn calf serum (NCS, Wisent BioProducts, Canada) and penicillin-streptomycin (PS, Wisent BioProducts, Canada). Two days prior to measurements, cells were transfected using polyethyleneimine (PEI, Polysciences Inc., Canada, No. 23966), with a ratio between PEI and DNA of 3:1. Afterwards, 20000 cells per well were seeded into white Cellstar PS 96-well cell culture plates (Greiner Bio-One, Germany). On the day of the measurement, medium was removed and Tyrode's buffer (137 mM NaCl, 0.9 mM KCl, 1 mM $MgCl_2$, 11.9 mM $NaHCO_3$, 3.6 mM $NaH_2PO_4$, 25 mM Hepes, 5.5 mM glucose, 1 mM $CaCl_2$, pH 7.4) was added, followed by incubation at 37 °C of at least 30 min. Ten minutes before measurement, adrenaline was added, with concentrations ranging from 31.6 nM to 3.16 mM in half-log steps, as well as a buffer control. At 5 min prior to measurement, coelenterazine 400a (DeepBlueC, Nanolight Technology) was added for a final concentration of 5 µM. Coelenterazine 400a was initially dissolved in DMSO and diluted into Tyrode's buffer with 1% Pluronic F-127 for increased solubility. BRET was measured in a Synergy Neo microplate reader (Biotek) using detection at 410 nm and 515 nm. All experiments were done at least in biological triplicates. Cut-off areas shown in Fig. 4 were calculated in a manner similar to the $M_3R$. Data analysis was done with RStudio 2021.09.2+382 (utilising R 4.1.2 and packages tidyverse 1.3.1 and drc 3.0-1).

**Calculation of residue contact maps**. For all class A and class B1 structures, residue contact maps were calculated only considering the transmembrane portions. A contact was defined to occur when the distance between any two atoms of two distinct residues was smaller than the sum of their van der Waals radii plus a buffer distance of 0.5 Å. In order to prevent sampling of local contacts which might introduce noise at later stages of the analysis, contacts between residues less than four positions apart in sequence and where one of the atoms involved was in the backbone were not considered.

**Creation of contact fingerprints**. In order to describe the residue contact distribution of each GPCR in a simplistic manner, a class-specific contact fingerprint was calculated for every structure. First, for class A and B1 GPCRs, the set of all residue-residue contacts that occurred in at least one of its members was compiled.

Only residues that were found in all analysed structures were considered. Further, this set of contacts was treated as a fingerprint in which the individual bits were set to either 1 or 0 depending whether or not a particular contact occurred in a structure. Finally, for each of our 557 structures, the appropriate class-specific fingerprint was calculated, the aggregate of which was then used to determine activation networks.

**Principal component analysis of fingerprints.** A class-specific principal component analysis was carried out based on the contact fingerprints using the python package `Scikit-learn` (version 0.23.2)[63]. Here, each structure can be seen as a sample while each contact can be seen as a variable. The contribution of each of the first 10 principal components to the overall variance of the data was plotted and evaluated. Then, PCA plots were created for the principal components that explained most of the variance. The data points, each of them representing one PDB structure, were coloured according to their activation state as ascribed in the GPCRdb. Principal components that showed a clear separation between the active and the inactive state were used in order to classify contacts important for the respective state. For each contact, the sign and the absolute value of the pertaining principal component coefficient gave the necessary information about the state and importance, respectively.

The procedures described above and the PCA were carried out again on those structures to which a clear inactive or active state could be assigned in the first PCA. This ensured that no intermediate structures were considered for the following analysis. For the class A PCA, we decided on structures with PC1 values larger than 7 and PC2 values larger than 7.5 for the inactive and active state, respectively. For class B1, structures with PC1 values larger than 3 were not considered for the re-calculation. Since the re-calculated class B1 PCA still showed two outliers, we decided on eliminating them (PDB: 6NIY https://doi.org/10.2210/pdb6NIY/pdb https://doi.org/10.2210/pdb6P9X/pdb) from the PCA before continuing with the network analysis.

**Contact analysis.** For both classes, the PCA coefficients were used in order to estimate the importance of a contact for a certain receptor state. Here, we focused on those contacts that were formed between residues in KS2, KS5, KS8, OS5, OS6 and OS9. Contacts between two residues that belong to the same helix were not considered. By investigating the re-calculated PCA plots (Fig. 2 and Supplementary Fig. 2), each contact was considered either as an active or inactive contact depending on the sign of its corresponding PCA coefficient. The PCA coefficients were normalised to their highest absolute value such that they ranged from 0 (not important) to ±1 (important). For each pocket, the residue contacts together with their normalised PCA coefficient were plotted.

**Reporting summary.** Further information on research design is available in the Nature Research Reporting Summary linked to this article.

## Data availability

The list of the 557 structures, aligned receptor coordinate files, and the probe docking data generated in this study are provided as Supplementary Data 1. Separate pymol sessions of the pocket densities for each class are provided as Supplementary Data 2. Both these datasets are also deposited in the Zenodo database under accession code https://doi.org/10.5281/zenodo.5973911. The experimental receptor activation data generated in this study are provided in the Supplementary Information and as Source Data in the source_data.xlsx file.

## Code availability

The structure retrieval pipeline is available at https://doi.org/10.5281/zenodo.5939894. SEED 4.1.2 is available from http://www.biochem-caflisch.uzh.ch/download. The code for the volumetric averaging software is available at https://github.com/torbengutermuth/volumetricaveraging. The software for primer design is located at https://github.com/dmitryveprintsev/AAScan.

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

## Acknowledgements

We thank Daniel Hilger and Anthony Watts for their helpful discussions. We thank the German Research Foundation DFG for Heisenberg Professorship KO4095/5-1 (to P.K.). The doctoral theses of J.B.H., M.P., and K.B. are funded by the LOEWE project "GLUE" (GPCR Ligands for Underexplored Epitopes; to M.B. (Moritz Bünemann) and P.K.) of the Hessen State Ministry of Higher Education, Research and the Arts. Marie Skłodowska-Curie Individual Fellowship from the European Union's Horizon 2020 research and innovation programme under grant agreement No. 844622 and American Heart Association's Grant #19POST34380839 (to F.M.H.). M.B. (Michel Bouvier) is funded by a Canadian Institute of Health Research Foundation Grant # 148431 (to M.B.) and holds the Canada Research Chair in Signal Transduction and Molecular Pharma-cology. F.M.H., M.B. (Moritz Bünemann), and P.K. are members of COST Action CA18133 "ERNEST" (European Research NEtwork on Signal Transduction).

## Author contributions

J.B.H., M.P., F.M.H., T.G., M.B. (Moritz Bünemann) and P.K. designed research; J.B.H., M.P., K.B., and F.M.H. performed calculations and experiments; J.B.H., M.P., K.B., F.M.H., M.B. (Michel Bouvier), M.B. (Moritz Bünemann) and P.K. analysed data; F.M.H., M.B. (Michel Bouvier), M.B. (Moritz Bünemann) and P.K. acquired funding; M.B. (Michel Bouvier), M.B. (Moritz Bünemann) and P.K. supervised research; all authors wrote the manuscript.

## Funding

## Competing interests

M.B. is the president of the scientific advisory board of Domain Therapeutics, a bio-technology company which licences BRET-based biosensors for commercial use. The biosensors used in the present study are freely available for academic research through material transfer agreements. The remaining authors have no conflicts of interest.
