## [Peer Review File · Nature Communications]

The pocketome of G-protein-coupled receptors reveals previously untargeted allosteric sitesREVIEWER COMMENTS

Reviewer #1 (Remarks to the Author):

This is an interesting and innovative manuscript. It defines in a rigorous way, using novel computational methods, binding cavities (GPCR pocketome) located both at the extrahelical part of GPCRs, facing the membrane bilayer, and within the 7TM bundle. These pockets are further classified as known (KS) or orphan (OS) sites, depending on whether known ligands bind to them or not, respectively, in crystal or cryo-EM structures. I would predict this will be an article of use for the wide community of medicinal chemists and especially the large GPCR research field, given the importance of GPCRs for signaling and drug discovery. I, thus, think the manuscript is suitable for publication if the authors provide minor revisions of their data:

1) The definition of the orthosteric (ORTHO) and adjacent secondary binding pockets (ORTHO1 and ORTHO2) for all GPCR classes is very confusing. The orthosteric binding cavities are mainly within the 7TM bundle in class A, the ECD and 7TM bundle in class B, and exclusively in the large ECD in classes C and F. Thus, sentences such as “the average orthosteric pocket for class C receptors is significantly deeper than for the one of class B1, which again is slightly deeper than the one in class A” or “the orthosteric pocket of class C extends down into this region [the sodium binding pocket (SODIUM)]” seem incorrect. The authors are probably comparing orthosteric in class A with allosteric binding sites in the other classes. I would suggest renaming the pockets according to the classes (combine ORTHO with ALLO) to avoid confusion. Moreover, secondary binding pockets in class A (ORTHO1 and ORTHO2) are not orthosteric and have been named, in addition to secondary, metastable binding sites or exosites.

2) Class C receptors form obligate cysteine-linked homodimers. The recently published inactive and active structures of these members show extensive contacts between TM helices of the dimer. Please clarify if the outward-facing cavities of class C receptors have been calculated in a single protomer or in the physiological dimer. It would be interesting to comment on whether ligands bound to these outward-facing cavities could block (NAMs) or assist (PAMs) protomer organization and, thus, class C receptor (in)activation.

3) I find difficult that ligands that bind on the outward-facing cavities can activate the receptor. Thus, it would be interesting to specify in Table 1 if the ligands are agonists or antagonists (inverse agonists) or PAMs or NAMs in addition to the location.

4) Figures 3 and 4 are very interesting. However, I have two concerns. First, quadruple mutants to only obtain a shift in EC50 for Gq recruitment of 1.5 or 2.25 times seems small. Second, these mutations in Figures 3 and 4 are performed in the key TM 5 and TM 6 and TM 7 and intracellular helix 8, involved in G protein binding. These mutations probably have indirect effects with key side chains involved in receptor activation, as the authors point. Therefore, these mutations do not show that ligand binding to OS1, OS2, KS1, KS2, KS3, ... , located in other less important helices will have similar effect. This should be clarify in the manuscript.

Reviewer #2 (Remarks to the Author):

Within their manuscript „The pocketome of G protein-coupled receptors reveals previously untargeted allosteric sites”, the authors present sound data, based on a carefully conducted study, by combining theoretical with lab-experimental techniques.

The authors could newly identify several sites within GPCRs, which were not addressed before. This might be important in development of new, more effective and more selective drugs. However, there are some revisions, I would ask for:

- 1) Page 2, lines 23-25: please mention at least one example
- 2) Page 4, line 59: The authors state that the probes, they used, may be too small to bind strongly to the receptor; if those probes are not binding strong, how can they effect the receptor activation (page 3, line 51/52)? Please clarify
- 3) It is well known that water molecules may bind in small pockets of GPCRs; how do the authors deal with this fact? at least they should include this in their discussion
- 4) Furthermore it is well known that GPCRs shown an dynamic behavior in the cell membrane and are not rigid; such dynamics may influence the structures of the small sites, the authors describe; how do the authors deal with this fact? At least they should include this in their discussion
- 5) Page 14, Figure 3: the authors specify a “normalized logEC50” in their figure; whereon was normalized? The same is true for Figure 4 on page 15
- 6) Page 15, line 241: The logical link between the small probes and the mutation studies is not quite clear; mutations on that amino acids may lead to change in the interaction network within the receptor, but this is no proof, that the sites, being in neighborhood to the mutation sites; please clarify
- 7) Within Figure S1 the authors mention the small probes, they used within their study; as the authors state by themselves, these probes might be too small to bind strongly to the pocket; so, why did the authors not include combinations of the probes, combined by an appropriate linker into their study?

8) It is not clear, if the authors propose that the small molecules bind individually into the newly identified binding sites, or if they have to be connected to another ligand

Response to reviewers for the manuscript “The pocketome of G protein-coupled receptors reveals previously untargeted allosteric sites” (NCOMMS-21-37924A)

Dear Reviewers,

We would like to express our gratitude to you for your responses and voicing your support and suggestions for improvement. These comments were very helpful for us.

Both of you viewed our manuscript very favourably, suggesting only minor revisions. We have carefully evaluated all the points raised, adapted the manuscript in multiple places, and provided additional explanations where requested. In the following, we will provide point-by-point responses to all your concerns. We are hopeful that we have found satisfactory answers that will lead to an acceptance of the manuscript.

Reviewer 1

We are grateful to the reviewer for their approval and the useful remarks that they provided.

1. The definition of the orthosteric (ORTHO) and adjacent secondary binding pockets (ORTHO1 and ORTHO2) for all GPCR classes is very confusing. The orthosteric binding cavities are mainly within the 7TM bundle in class A, the ECD and 7TM bundle in class B, and exclusively in the large ECD in classes C and F. Thus, sentences such as “the average orthosteric pocket for class C receptors is significantly deeper than for the one of class B1, which again is slightly deeper than the one in class A” or “the orthosteric pocket of class C extends down into this region [the sodium binding pocket (SODIUM)]” seem incorrect. The authors are probably comparing orthosteric in class A with allosteric binding sites in the other classes. I would suggest renaming the pockets according to the classes (combine ORTHO with ALLO) to avoid confusion. Moreover, secondary binding pockets in class A (ORTHO1 and ORTHO2) are not orthosteric and have been named, in addition to secondary, metastable binding sites orexosites.

Response: *We agree that this might have caused confusion and have now renamed ORTHO, ORTHO1, and ORTHO2 to IBS1 (for interhelical binding site 1), IBS2, and IBS3, respectively, in order to ameliorate this point. There is changed text on page 5 which also mentions this in more detail: “In each class, a large interhelical site (Interhelical Binding Site 1, IBS1) and adjacent sec-*

ondary binding pockets (IBS2 and IBS3) can clearly be discerned. Whereas IBS1 represents the classic orthosteric site in class A GPCRs, it forms – together with the extra-cellular domain (ECD) – the peptide binding site in class B GPCRs. Furthermore, IBS2 and IBS3 are two known exosites in class A GPCRs. Since the orthosteric site of class C receptors is located in the ex- tracellular Venus flytrap (VFT) domain, IBS1 is commonly referred to as an allosteric site in class C receptors. Our methodology was able to correctly depict the size and shape of these known pockets for different classes, and we therefore hypothesized that the other pockets identified in this work can indeed also host ligands. By aligning our density maps with each other, one can see that the average IBS1 for class C receptors protrudes significantly deeper than the one of class B1, which again goes slightly deeper than the one in class A. This is perfectly consistent with experimental evidence.” We hope that the reviewer agrees with this altered nomenclature.

2. Class C receptors form obligate cysteine-linked homodimers. The recently published inactive and active structures of these members show extensive contacts between TM helices of the dimer. Please clarify if the outward-facing cavities of class C receptors have been calculated in a single protomer or in the physiological dimer. It would be interesting to comment on whether ligands bound to these outward-facing cavities could block (NAMs) or assist (PAMs) protomer organization and, thus, class C receptor (in)activation.

Response: Thank you for bringing up this interesting point. We have not taken into account dimers, which we now state more explicitly on page 4: “Please note that, for our approach, we did not consider dimerisation of the 7TM bundle (as has been described for class C GPCRs), but rather docked to the individual monomers. Moreover, we treated each receptor structure as rigid. We provide a list of all analysed structures together with additional information as a supplementary [.csv] file.” and in the Methods section on page 23: “For each structure, the transmembrane portion and adjacent motifs belonging to the GPCR were separated from all non-native insertions (i.e. non-GPCR proteins, water, other small molecules) by using the information about the preferred chain retrieved from the GPCRdb and the DBREF tag in the PDB file. In the case of dimeric GPCR structures, only one of the monomeric subunits was considered for docking.”. However, the pockets that we identified are indeed close to the relevant regions of class C receptors, and we added text saying so on page 7: “Despite the fact that we only considered monomeric subunits of the 7TM bundle in our calculations, our methodology was able to also reveal all dimerisation interfaces, which have predominantly been described for class C GPCRs. The conserved TM6-TM6 dimerisation interface in active-state class C receptors encompasses KS7, KS8, KS9, and partially OS5 and is known to bind positive allosteric modulators (PAMs).^{12,13} Two other dimerisation interfaces can be found between TM3-TM4 (mGlu2) or TM3-TM5 (GABA B) in inactive-state class C GPCRs.^{14,15} While the former is mainly formed by residues at the extracellular end of the helices and is thus represented by KS2, the latter dimerisation interface is located in the region of KS5.”. Whether or not ligands binding at these sites will predominantly act as PAMs or NAMs is beyond what we can say based on our docking calculations, however.

3. I find difficult that ligands that bind on the outward-facing cavities can activate the receptor. Thus, it would be interesting to specify in Table 1 if the ligands are agonists or antagonists (inverse agonists) or PAMs or NAMs in addition to the location.

Response: We gladly provide this information by adding a tag which specifies the nature of the modulation that each ligand causes to each PDB ID in Table 1. This data has been lifted from GPCRdb.

4. Figures 3 and 4 are very interesting. However, I have two concerns. First, quadruple mutants to only obtain a shift in EC50 for Gq recruitment of 1.5 or 2.25 times seems small. Second, these mutations in Figures 3 and 4 are performed in the key TM 5 and TM 6 and TM 7 and intracellular helix 8, involved in G protein binding. These mutations probably have indirect effects with key

side chains involved in receptor activation, as the authors point. Therefore, these mutations do not show that ligand binding to OS1, OS2, KS1, KS2, KS3, . . . , located in other less important helices will have similar effect. This should be clarify in the manuscript.

Response: *Thank you for raising this issue. First, we would like to point out that in the left panel of Figures 3 & 4, the values on the x-axis are on a log scale, and therefore represent more than 10-fold change. We realised that our use of the word “normalisation” was misleading and therefore adapted the wording in figure and text (cf. also our answer to reviewer 2, question 5). Second, we chose residues such that their side chains point inside the pockets and not towards the G protein site. We have added a new Figure S9 that illustrates the location of these residues and have adapted the text to now read: “In order to identify the impact of OS5 and OS9 on the biological function of the receptors, we also conducted mutation studies with the M₃R and β₂AR. A visualisation of location of the residues mutated in our experiments is shown in Fig. S9. Of note, we chose the residues such that their side chains are pointing into the sites, and are thus available for interaction with a ligand. For the M₃R, we constructed double or quadruple mutants, where two or four, respectively, of the residues that form these pockets were changed (residues mutated in a particular mutant are connected by a grey vertical bar in Fig. 3).” Of course, we cannot say anything about the other orphan sites or guarantee that OS5 or OS9 will be relevant sites to target in all GPCRs. However, for the known sites for which ligands exist which have led to altered receptor responses, we think that it is safe to assume that they can be exploited for small-molecule modulators. To clarify this, we have adapted the last paragraph on page 22: “Of course, in these cases, the challenges will be to first demonstrate that each orphan site (beyond OS5 and OS9) can indeed be exploited to modulate receptor function by small-molecule ligands, to unequivocally. . .”.*

Reviewer 2

We thank the reviewer for the overall support of our work and the careful reading of the text.

1. Page 2, lines 23-25: please mention at least one example

Response: *We have gladly added one example for each medication, i.e. bisoprolol and salmeterol, respectively, as well as a citation for a comprehensive review on this topic (Baker & Wilcox 2017).*

2. Page 4, line 59: The authors state that the probes, they used, may be too small to bind strongly to the receptor; if those probes are not binding strong, how can they effect the receptor activation (page 3, line 51/52)? Please clarify

Response: *The probes were not intended as ligands on their own, but just to investigate the details of the pockets. We have tried to clarify this further by adding “. . . after stimulation with an orthosteric agonist” on page 3, line 53 and “. . . and are thus suited to investigate the details of cavities on receptors” on page 4, line 62.*

3. It is well known that water molecules may bind in small pockets of GPCRs; how do the authors deal with this fact? at least they should include this in their discussion

Response: *This is indeed true. However, we found only very few water molecules resolved in the transmembrane pockets. We better explained this point on page 20: “Most of the sites identified in this work are located on the outward-facing portion of the 7TM bundle within the membrane, and water molecules close to an allosteric ligand were only observed in 14 of the 71 structures that featured an allosteric ligand and therefore not investigated further. It cannot be ruled out that they play a role in certain sites or for certain ligands, but this is likely more important for sites that are directly accessible by the solvent such as the G protein-binding site or KS11.”.*

4. Furthermore it is well known that GPCRs shown an dynamic behavior in the cell membrane and are not rigid; such dynamics may influence the structures of the small sites, the authors describe;

how do the authors deal with this fact? At least they should include this in their discussion

Response: *An excellent point, which we have indeed not clarified enough in the previous version. We have included this fact also in the text provided in response to question 2 of reviewer 1 and added two sentences on page 20: "Of course, our analysis is based on rigid receptor structures and it stands to reason that some of these sites will undergo rearrangements upon changes of receptor conformation. This will need appropriate attention during ligand design."*

5. Page 14, Figure 3: the authors specify a "normalized logEC50" in their figure; whereon was normalized? The same is true for Figure 4 on page 15

Response: *We apologise for this use of jargon. The left panel actually depicts a difference between the logEC₅₀ values. We have updated the figures, and added a more complete definition in the caption: "The left panel shows the difference between the mutant logEC₅₀ and the mean wt logEC₅₀ values ($\log EC_{50}^{mut} - \log EC_{50}^{wt}$) and the right panel the normalised amplitude (Amp^{mut}/Amp^{wt}) of the extent to which β -arrestin2 was recruited."*

6. Page 15, line 241: The logical link between the small probes and the mutation studies is not quite clear; mutations on that amino acids may lead to change in the interaction network within the receptor, but this is no proof, that the sites, being in neighborhood to the mutation sites; please clarify

Response: *This point was also raised by reviewer 1, and we ask this reviewer to kindly refer to our answer to their question 4.*

7. Within Figure S1 the authors mention the small probes, they used within their study; as the authors state by themselves, these probes might be too small to bind strongly to the pocket; so, why did the authors not include combinations of the probes, combined by an appropriate linker into their study?

Response: *We thank the reviewer for pointing out this approach. We have therefore added two sentences on page 20: "As mentioned before, the probes by themselves are likely too small to bind with reasonable affinity. However, one can certainly imagine that connections of individual probes with appropriate linkers will lead to higher-affinity ligands, a future direction of research."*

8. It is not clear, if the authors propose that the small molecules bind individually into the newly identified binding sites, or if the they have to be connected to another ligand

Response: *The probes we used are not intended as ligands by themselves (although they might be connected with each other or a bigger ligand, cf. our response to your previous question). Thus, we believe that a screen for lead-like-sized ligands holds the greatest promise at the moment. We hope that the many textual changes we introduced in the new version of the manuscript made this point clearer.*

As stated at the beginning, we hope that we provided satisfactory answers to all issues raised and would be delighted to see our work published in *Nat. Commun.*

Sincerely,

Peter Kolb
Professor of Pharmaceutical Chemistry

REVIEWERS' COMMENTS

Reviewer #1 (Remarks to the Author):

The authors have included all my suggestions in this revised version. It is suitable for publication as it is.

Reviewer #2 (Remarks to the Author):

The authors have addressed all items in an appropriate manner. In my opinion no further changes are needed.